# Spatiotemporal profiling of cytosolic signaling complexes in living cells by selective proximity proteomics

Mi Ke[1,6], Xiao Yuan[1,6], An He[1], Peiyuan Yu [1], Wendong Chen[1], Yu Shi[2], Tony Hunter [2], Peng Zou [3] & Ruijun Tian [1,4,5]✉

Signaling complexes are often organized in a spatiotemporal manner and on a minute timescale. Proximity labeling based on engineered ascorbate peroxidase APEX2 pioneered in situ capture of spatiotemporal membrane protein complexes in living cells, but its application to cytosolic proteins remains limited due to the high labeling background. Here, we develop proximity labeling probes with increased labeling selectivity. These probes, in combination with label-free quantitative proteomics, allow exploring cytosolic protein assemblies such as phosphotyrosine-mediated protein complexes formed in response to minute-scale EGF stimulation. As proof-of-concept, we systematically profile the spatio-temporal interactome of the EGFR signaling component STS1. For STS1 core complexes, our proximity proteomics approach shows comparable performance to affinity purification-mass spectrometry-based temporal interactome profiling, while also capturing additional—especially endosomally-located—protein complexes. In summary, we provide a generic approach for exploring the interactome of mobile cytosolic proteins in living cells at a temporal resolution of minutes.

---

[1] Department of Chemistry, School of Science, Southern University of Science and Technology, Shenzhen, China. [2] Molecular and Cell Biology Laboratory, Salk Institute for Biological Studies, La Jolla, CA, USA. [3] College of Chemistry and Molecular Engineering, Peking University, Beijing, China. [4] Guangdong Provincial Key Laboratory of Cell Microenvironment and Disease Research, Southern University of Science and Technology, Shenzhen, China. [5] Shenzhen Grubbs Institute, Southern University of Science and Technology, 518055 Shenzhen, China. [6] These authors contributed equally: Mi Ke, Xiao Yuan. ✉email: tianrj@sustech.edu.cn

Protein machines are assembled and spatiotemporally controlled by protein complexes in a sophisticated manner[1,2]. In signaling network mediated by tyrosine phosphorylation (pTyr), for example, pTyr sites are precisely regulated by tyrosine kinases and tyrosine phosphatases on the minute timescale[3], while pTyr signaling complexes are dynamically organized by recognizing pTyr sites through specific binding domains, including Src homology 2 (SH2) and pTyr-binding (PTB) domains. With recent advances in affinity purification combined with mass spectrometry (AP-MS) and various quantitative proteomics approaches, dynamic pTyr signaling complexes have been well characterized[4–7]. However, affinity purification-based methods after cell lysis lead to the loss of spatial information and weak interactions, limiting their application in the unbiased discovery of spatiotemporal signaling complexes with biological significance.

With its unique feature of labeling neighboring proteins within an ~10 nm radius in living cells, proximity-dependent biotinylation has recently been adopted for studying protein complexes on a proteome scale[8]. BioID was first introduced and has been widely applied for capturing stable protein complexes in living cells by tagging the lysine residues of a proximal protein with biotin[9–11]. However, BioID requires ~12–24 h to gain enough labeling signals, which limits its application for studying temporal protein complexes. TurboID was recently evolved from BioID to achieve biotin labeling in 10 min, making it possible to study more dynamic protein complexes[12]. In a different approach, ascorbate peroxidase APEX was engineered and extensively applied to study the subproteome in various subcellular structures[13,14]. Compared with BioID and TurboID, APEX generates biotin-phenol radicals upon activation with $H_2O_2$ and achieves protein labeling in <1 min. Taking advantage of its fast labeling kinetics, two pioneering studies have demonstrated the potential application of APEX-based proximity labeling for exploring dynamic GPCR signaling complexes with subminute resolution and even subcellular translocalization[15–17]. Although background labeling introduced by the highly reactive biotin-phenol cloud could be largely distinguished by tagging membrane-localized GPCRs and using quantitative proteomics, it is still a major challenge for the general application of APEX-based proximity proteomics approach to explore dynamic protein complexes with spatiotemporal resolution, especially for cytosolic signaling proteins.

It has been predicted that the labeling radius of the biotin-phenol radical and therefore the labeling specificity of APEX-based proximity labeling could be modulated by modifying the aromatic ring with chemical substituents, and this approach has confirmed the robust proximity labeling of subproteome by biotin-phenol and leaded to the recent development of highly efficient APEX-based RNA labeling[13,18]. In this study, we design a series of biotin-phenol analogs by modifying the phenolic hydroxyl structure. We obtain BP5 and BN2, which show higher labeling specificity for protein complexes than biotin-phenol. Based on label-free quantitative proteomics (LFQ), we apply BP5- and BN2-based proximity proteomics to explore pTyr adapter protein complexes and other stable protein complexes in cytosol. We further identify and systemically characterize the temporal interactome of an EGFR signaling component STS1 (TULA-2 or UBASH3B) with minute-scale temporal resolution. Side-by-side comparison with AP-MS data generated with the same cell line confirms the comparable performance for profiling temporal core EGFR signaling complexes and demonstrates the potential of the proximity proteomics approach for exploring many more spatial protein complexes including RABEP1 located at the endosome. These results demonstrate the utility of biotin-phenol analogs BP5- and BN2-based APEX2 labeling and label-free quantitative proteomics for exploring cytosolic protein complexes in living cells with spatiotemporal resolution.

## Results

**Design and synthesis of biotin-phenol analogs with high reactivity**. Mechanistically, APEX-mediated proximity labeling involves APEX-catalyzed conversion of phenolic hydroxyls of both the biotin-phenol and tyrosine residues in neighboring proteins into free radicals. The free radicals then form covalent bonds that result in the biotinylation of proteins, which can then be enriched and analyzed by MS. To reduce the labeling radius of biotin-phenol radicals and thereby improve the labeling selectivity, we designed and synthesized 12 biotin-phenol analogs with chemical modifications on the phenolic hydroxyl, including biotin-phenol (BP1), BP2, BP3, BP8, and BP9, which have been described previously[13]; BP4, BP5, BP6, BP7, and BP10 which were newly synthesized by this study (Fig. 1a). In addition, we also synthesized BN1 and BN2 with aromatic amine structures as reported recently by us with weak protein labeling activity[18]. The general rationale for selecting these probes is that the bond dissociation energy (BDE) of the -OH bond and $-NH_2$ bond of the benzene ring could be modulated by introducing chemical substituent[13,19,20]. The collection of biotin-phenol analogs therefore represents enough diversity with six of them modifying the para-position, three of them modifying the ortho-position and two of them containing the aromatic amine structures.

We first compared the activity of each biotin-phenol analog for forming free radicals through catalysis by horseradish peroxidase (HRP) plus $H_2O_2$ in vitro and APEX2 plus $H_2O_2$ in living cells (Fig. 1b). We observed that the consumption rate of BP1 was ~61% after 20 min of reaction. Interestingly, BP5, BP10, and BN2 had higher consumption rate than BP1 in which BP5 and BP10 reached to ~95% and BN2 reached to ~65%, respectively. In comparison, other probes showed reactivity lower than 50% or almost no reactivity (Fig. 1c and Supplementary Fig. 1a). In addition, when we investigated the products of the biotin-phenol analogs by LC-MS analysis, we not only found dimers for BP1, BP2, BP5, BP6, BP7, BP8, and BN2, but also surprisingly found a large amount of BP5 trimer, which should indicate their higher reactivity (Supplementary Fig. 1b and Supplementary Data 1). Furthermore, consistent with recent report for synthesis of aniline-based polymers catalyzed by APEX2 in living cells[21,22], we found BN2 could efficiently form red-colored polymers (Supplementary Fig. 1c). We went on to investigate BP5, BP10, and BN2 with higher reactivity than BP1 using stably expressed cytosolic APEX2 in HeLa cells. As shown in Fig. 1d, BP5 and BN2 have lower labeling intensity than BP1 as indicated by streptavidin western blot. The lower labeling intensity of BN2 is consistent with our recent report[18]. Unexpectedly, BP10 has almost no reactivity in living cells for unknown reasons and is therefore discarded for further investigation.

The high reaction activity of BP5 suggests that it is likely to generate free radicals efficiently. To test this hypothesis and make direct comparisons between BP1 and BP5, we monitored the reaction dynamics at 0, 1, and 20 min by mixing BP1 and BP5 together with HRP and $H_2O_2$ in one reaction tube or separately. We observed a decrease in the consumption rate of BP1 from ~60% to ~20% when mixed with BP5, while the consumption rate of BP5 was not affected by BP1 and remained at ~95% in both conditions (Fig. 1e). Moreover, we observed that the time point of BP1 dimer production was delayed to 20 min, whereas it reached to 60% at 1 min when reacting with HRP and $H_2O_2$ alone. However, the production rates of BP1–BP5, BP5 dimer, and BP5 trimer were barely affected when BP1 was mixed together (Supplementary Fig. 1d). We also made comparison for BP5 vs. BN2 with the same in vitro reaction design and confirmed the higher in vitro reactivity of BP5 (Supplementary Fig. 1e). To further mimic the real reaction in the cells, we added tyrosine to the reaction system. As expected, the production of BP1-tyrosine

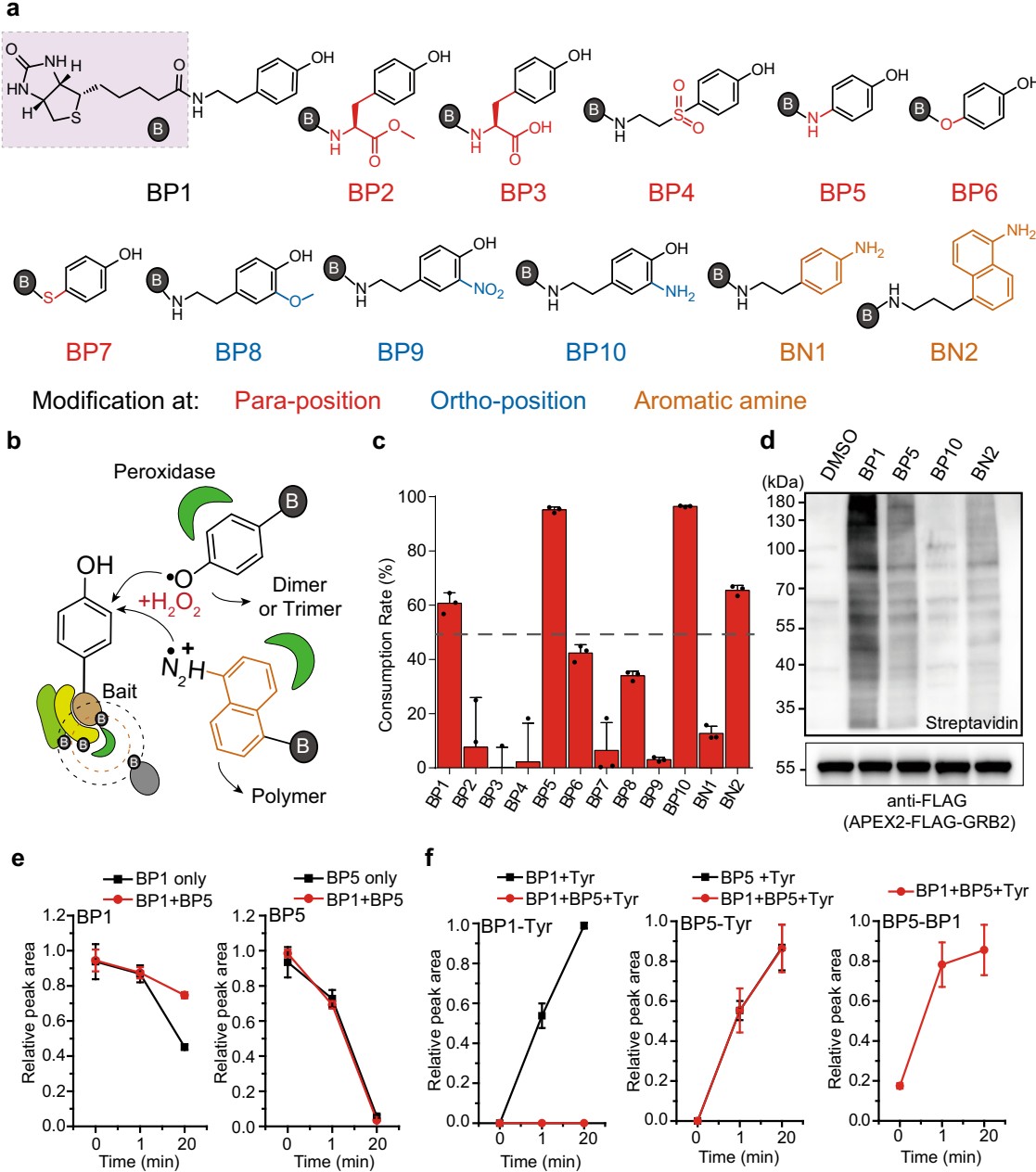

**Fig. 1 Design and synthesis of biotin-phenol (BP) derivatives with high reactivity. a** Design of different BP derivatives with substituents at para-position (red), ortho-position (blue), and with aromatic amine structures (orange). **b** Scheme of proximity labeling catalyzed by peroxidases in living cells and in vitro. **c** Consumption rate of BP derivatives in the presence of 10 nM HRP and 500 µM $H_2O_2$ for 20 min. Each dot represents the indicated data of one independent experiment. Data are presented as mean values ± standard divations (s.d.; $n = 3$ independent biological experiments). **d** Streptavidin western blot analysis of labeling activity of BP1, BP5, BP10, and BN2 in living HeLa cells with stably expressed APEX2-FLAG-GRB2 fusion protein ($n = 3$ independent biological experiments). After incubating the cells with probes for 30 min, 500 µM of $H_2O_2$ was added for 1 min reaction. Quantification was presented in Supplementary Fig. 12a. **e, f** Comparison between BP1 and BP5 reactivity with or without tyrosine when incubated together. Data are presented as mean values ± s.d. Error bars in **c**, **e**, and **f** represent s.d. as quantified by LC-MS ($n = 3$ independent biological experiments). Source data are provided as a Source Data file.

was greatly diminished by BP5 (almost no BP1-tyrosine production), while the generation of BP5-tyrosine was not affected in the presence of BP1 (Fig. 1f and Supplementary Fig. 1f). Lastly, we confirmed in vitro by using FLAG peptide and pTyr peptide derived from CD28 protein that both BP5 and BN2 probes preferentially labeled tyrosine without affect to the phosphorylated tyrosine on the same peptide sequence (Supplementary Fig. 1g–j). To further support above experimental observations, we also performed density functional theory (DFT)

computations for evaluating the BDE of key functional group of BP1 or tyrosine, BP4, BP5, BN1, and BN2, and confirmed the lowest BDE of BP5 which is favorable for generating radicals (Supplementary Fig. 2 and Supplementary Data 2). In addition, DFT computations also confirmed the lowest reaction barrier of BN2 with tyrosine radical compared with BP5 and BP1. In summary, we successfully developed biotin-phenol analogs, BP5 and BN2, which are prone to conversion to free radicals in vitro in a faster manner and may interact with substrates in a more

selective manner in living cells. Consistent with recent discovery that quenching of highly reactive intermediates is critical for high-selective proximity labeling of extracellular protein complexes, selective biotinylation by BP5 and BN2 in living cells is reasonable as further labeling of substrate proteins with larger radius is not favorable due to the robust formation of BP5-dimer and trimer and BN2 polymers before diffusing farther out for labeling neighboring protein in a nonspecific manner[23].

**Development of a highly selective proximity proteomics approach**. To date, most of the proteins studied by the APEX approach have been transmembrane proteins or proteins located in compact subcellular structures, such as stress granules[24]. Here, we aimed to study the spatiotemporal interactome of cytosolic adapter proteins, as they often function as generic scaffolds for many signaling pathways[25]. The lower labeling intensity of BP5 and BN2 in living cells suggests their potential for selectively labeling surrounding proteins which form complex with the protein of interest. To test this hypothesis, we developed an inducible lentivirus infection system with puromycine selection for stably expressing the APEX2 enzyme fused to a gene of interest with a FLAG tag as a linker (Supplementary Fig. 3a)[26]. We first assembled the small adapter protein GRB2 that contains two SH3 domains and one central SH2 domain into the system, tuned its expression in HeLa cells close to the endogenous level by adjusting the doxycycline (Dox) concentration, and labeled it and associated proteins in living cells upon 1 min $H_2O_2$ treatment (Supplementary Fig. 3a, b). We were specifically interested in signaling complexes mediated by receptor tyrosine kinases (RTKs), such as EGFR, as they are typically activated and perform spatiotemporal regulation of downstream signaling protein complexes within minutes. Importantly, we did not observe distinguishable pTyr activation of HeLa cells upon treatment with 0.5 mM of $H_2O_2$ which is two times lower than the original application[13] (Supplementary Fig. 3c), demonstrating the neglectable artificial activation of pTyr by $H_2O_2$ (even though elevated $H_2O_2$ can inactivate tyrosine phosphatases and increase pTyr levels)[27].

We then tested whether BP5-based proximity labeling is suitable for interactome profiling with minute resolution. A previously developed label-free quantitative proteomic workflow was adopted for quantitative analysis of the temporal interactome[6]. After proximity labeling in living cells and cell lysis, streptavidin beads-based affinity purification, strong wash under denaturing condition and on-bead digestion were performed before label-free quantitative proteomic analysis (Fig. 2a). After 2 min of EGF stimulation, 59 and 23 proteins were reproducibly differentiated (biological triplicate) by BP5 and BP1 as stimulation-dependent GRB2-associated proteins (Fig. 2b, c, Supplementary Fig. 11a, and Supplementary Data 3). We selected S0 = 0.5 and FDR < 0.05 as cutoff according to the identification of known GRB2-interacting proteins with better differentiation of the real hits from background noise (Supplementary Fig. 3d, e). Interestingly, 37 of the BP5 unique proteins were found with no significant change in the background noise of the BP1 results, which demonstrates the better labeling selectivity of BP5. Functional annotation demonstrated that BP1 and BP5 behaved similarly for identifying core EGF stimulation-dependent GRB2 protein complexes with proximity to plasma membrane, but BP5 identified many more signaling proteins related to signal transduction, trafficking, etc. (Fig. 2d)[6,28,29]. GRB2 interactions with EGFR, SHC1, CBL, and STS1 upon EGF stimulation were further validated by western blot and fluorescence colocalization (Supplementary Fig. 3f–i). To additionally validate the successful biotinylation of GRB2-interacting proteins, we enriched and

identified BP5-modified peptides from EGF-treated HeLa cell lysate (Supplementary Fig. 4 and Supplementary Data 4). Among 650 BP5-modified peptides covering 435 proteins, GRB2 and 11 interacting proteins were confidently identified with BP5-modified peptides, including STS1, CD2AP, CRK, WIPF2, EZR, WASL, EIF3J, CBL, PTPN11, ANKS1A, and ZDHHC5. Furthermore, we compared the LFQ intensity of 26 known GRB2-interacting proteins identified by both BP5 and BP1 and found that 62% of these proteins were identified with higher intensity in the BP5 experiment than in BP1 (Fig. 2e). In comparison, 134 identified proteins with cytosolic location annotation by Gene Ontology (GO) were identified with higher frequency by the BP1 experiment (63%). This analysis further validated the high-selective labeling of protein complexes by BP5 rather than surrounding cytosolic proteins with larger radius, which is consistent with the result indicated in Fig. 1d. It is therefore conclusive that BP5 could effectively differentiate GRB2-interacting proteins from background noise in living cells and identified significantly more known interaction and functionally relevant proteins.

To further confirm the better labeling selectivity of BP5 against BP1 for profiling EGF stimulation-dependent interactome, we APEX2-tagged and proximity labeled two identified GRB2-interacting proteins, STS1 and SHC1. Consistent with the proximity labeling for GRB2, BP5 selectively labeled significantly more STS1- and SHC1-interacting proteins upon 2 min of EGF stimulation (Supplementary Figs. 5a–f and 11b, c). Additionally, we made comparison between BP5 and BN2 for selectively labeling EGF stimulation-dependent GRB2 complex. As shown in Supplementary Fig. 5g, h, Supplementary Fig. 11d, and Supplementary Data 5, BP5 outperformed BN2 for selectively labeling significantly more known GRB2-interacting proteins. The labeling intensity of BN2 for bait protein GRB2 is ~20 times higher than BP5 (Supplementary Fig. 5i). Additional analysis revealed that well characterized direct-associated proteins, such as EGFR and SHC1, have much higher labeling intensity as well, while other associated proteins, such as WIPF2, CD2AP, and EPS15, have lower labeling intensity than BP5 (Supplementary Fig. 5i). These results indicated that BN2 might have much smaller labeling radius for covering and robustly labeling directly associated proteins. This performance is expectable because that BN2 has the lowest reaction barrier for efficiently labeling neighboring proteins (Supplementary Fig. 2b) and could efficiently form polymers before diffusing farther out. It should be noted that we also compared the proximity labeling performance of BP5 with BP6 and BP8 which have 30–40% of consumption rates in vitro (Fig. 1c and Supplementary Fig. 5j). Expectedly, although BP8 could differentiate three known GRB2-interacting proteins from background noise, its performance is much worse than BP5 (Supplementary Fig. 5k, l and Supplementary Fig. 11e). Same as BP10, BP6 has no labeling activity in living cells for unexpected reason and was not pursued for further investigation.

It should be noted that EGF stimulation robustly recruits cytosolic GRB2 to plasma membrane-embedded EGFR, making the selective labeling of membrane-associated EGFR signaling complexes less challenging than the cytosolic proteins with free mobility. We went on to compare BP5-, BN2-, and BP1-based proximity proteomic analysis for ILK which locates in cytosol and is the core component of well-characterized IPP complex, including ILK, Parvins (PARVA and PARVB), and PINCH (Fig. 3a)[6,30]. Although BP5-based proximity labeling well differentiated bait protein ILK, PINCH, and RSU1 (direct-interacting protein of PINCH) from background noise and GFP as control (Fig. 3b), PARVA and PARVB were not identified. BP1-based proximity proteomic analysis of ILK got even worse

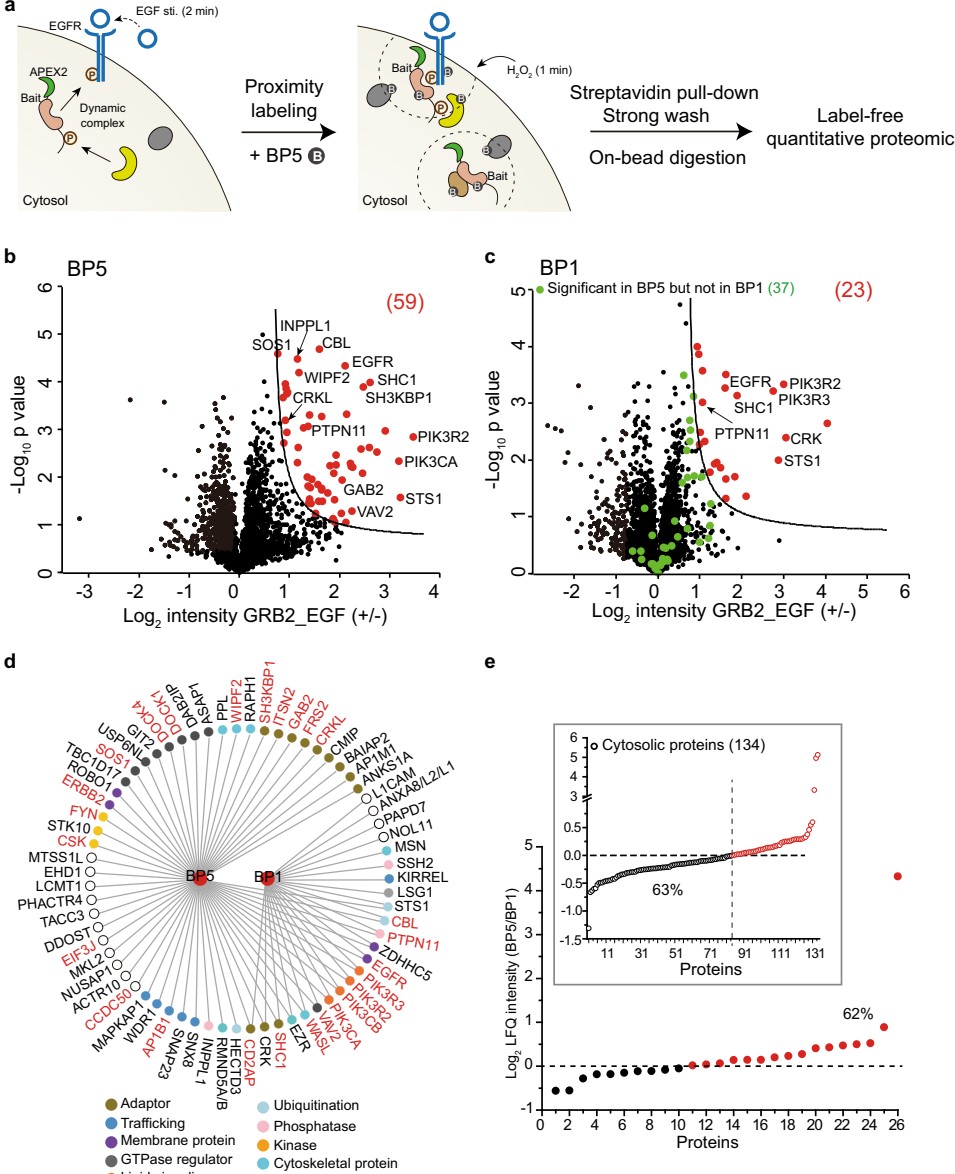

**Fig. 2 Development of a highly selective proximity proteomics approach for temporal membrane-associated and cytosolic interactome. a** Scheme of BP5-based proximity proteomic analysis of adapter protein interactome upon EGF stimulation in living cells. **b**, **c** Volcano plots of GRB2-interacting proteins in HeLa cells with stably expressed APEX2-FLAG-GRB2 quantified by BP5- and BP1-based proximity proteomics ($n = 3$ independent biological experiments). Significantly changed proteins upon 2 min EGF stimulation are highlighted in red (FDR < 0.05 and S0 = 0.5). Relevant proteins that are significant in BP5 experiment but not in BP1 experiment are marked in green. **d** Interaction map and function annotation of significant GRB2-associated proteins identified by BP5- and BP1-proximity proteomics. The colored circles indicate the functional classification. Known GRB2- associated proteins were labeled in red. **e** Comparison of BP5 and BP1 labeling selectivity as indicated by both known GRB2-associated proteins and cytosolic proteins with GO annotation of cytosol identified from the same pull-down sample (insert). The average ratio of LFQ intensity (BP5/BP1) from three replicates were plotted. Source data are provided as a Source Data file.

result with only RSU1 differentially labeled (Supplementary Fig. 6a). On the contrary, BN2-based proximity proteomic analysis precisely labeled and identified all the IPP complex proteins with much higher selectivity (Fig. 3c, Supplementary Fig. 11f, and Supplementary Data 6). The superior performance of BN2 was further confirmed by reciprocally labeling RSU1 and PINCH with even higher selectivity as noted by the log₂ intensity ratio (Fig. 3d, e, Supplementary Fig. 6b, c, Supplementary Fig. 11g, h, and Supplementary Data 7 and 8). Consistent with the observation above for BN2-based proximity labeling of GRB2 interactome (Supplementary Fig. 5h, i), all the five proteins in the

same complex demonstrated much higher labeling intensity by BN2 compared with BP5 (Fig. 3f and Supplementary Fig. 6d), which further confirmed BN2's preference for labeling direct-associated protein complex within short radius in living cells. In addition, background labeling by different probes as represented by GFP intensity further confirmed the high-selective labeling of BN2 compared with BP1 and BP5 (Supplementary Fig. 6e). Collectively, we successfully developed and explored two biotin-phenol analogs, BP5 and BN2, for labeling cytosolic protein complexes with significantly higher labeling selectivity than the original BP1. BN2 is suitable for profiling cytosolic protein

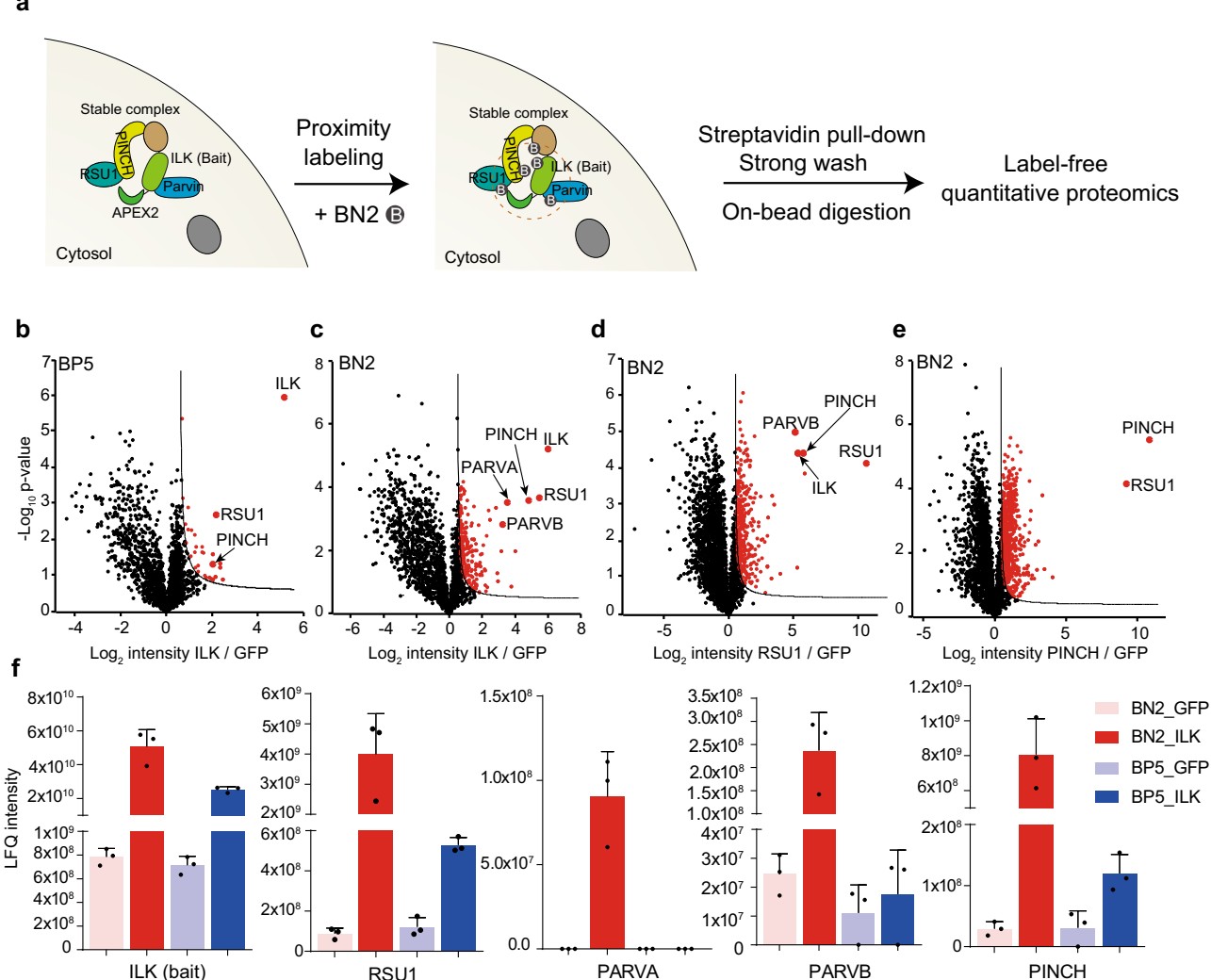

**Fig. 3 Highly selective proximity proteomic analysis of stable protein complex in cytosol. a** Scheme of BN2-based proximity proteomic analysis in living cells. The well-characterized stable ILK-PINCH-PARVIN-RSU1 protein complex are shown (insert). **b, c** Volcano plot comparison between BP5- and BN2-based proximity proteomic analysis with stably expressed APEX2-FLAG-ILK in HT1080 cells as the bait protein ($n = 3$ independent biological experiments). **d, e** Reciprocal BN2-based proximity proteomic analysis of RSU1 (**d**) or PINCH (**e**) interactome. Stably expressed GFP was used as control in b–e. Significantly changed proteins in b–e were highlighted in red (FDR < 0.05 and S0 = 0.5). **f** LFQ intensity comparison of the ILK-PINCH-PARVIN-RSU1 complex proteins identified by the BP5- and BN2-based proximity proteomic analysis with ILK as bait protein. Each dot represents the indicated data of one independent experiment. Data are presented as mean values ± s.d. Error bars represent s.d. ($n = 3$ independent biological experiments). Source data are provided as a Source Data file.

complex with direct association in smaller radius, while BP5 could efficiently label membrane-associated and cytosolic protein complexes in larger radius.

**Minute-resolution temporal profile of the STS1 interactome.** With the successful application of the highly selective BP5-based proximity proteomics approach for profiling EGF stimulation-dependent signaling complexes associated with adapter proteins GRB2, SHC1, and STS1, we further challenged the BP5-based proximity proteomic approach for exploring the temporal interactome of STS1 upon an EGF stimulation series in living cells, including 0, 2, 5, 10, and 30 min (Fig. 4a, b, Supplementary Figs. 7a and 11i, and Supplementary Data 9). STS1 contains multiple domains, including a pseudotyrosine phosphatase domain, a SH3 domain and an UBA domain, which is known to interact with and potentially dephosphorylate EGFR signaling complexes[31–33]. However, its interactome and dynamic

modulation in EGFR signaling pathway have not been characterized by MS-based proteomics. Taking advantage of 1 min labeling kinetics of BP5- and APEX2-based proximity labeling and LFQ-based quantitative proteomics, EGF stimulation-dependent STS1 interactome from each time point was precisely characterized with a medium CV of ~0.1 (Supplementary Fig. 7b).

To accurately chart the dynamic changes of the STS1 interactome across all five time points of EGF stimulation, we developed a data analysis pipeline and manually curated four highly correlated clusters for 35 identified STS1-interacting proteins (Fig. 4c, d and Supplementary Fig. 7c), including (1) cluster 1 with a peak at 2 min of EGF stimulation, in which most of the proteins are well-known EGFR signaling complex components, such as CBL, PTPN11, and SHC1 (Supplementary Fig. 7d); (2) cluster 2, which exhibited continuously increasing recruitment to STS1 and reached a peak at ~10 min, with GO annotation indicating enrichment of proteins related to subcellular organelles and structures; (3) cluster 3, which exhibited

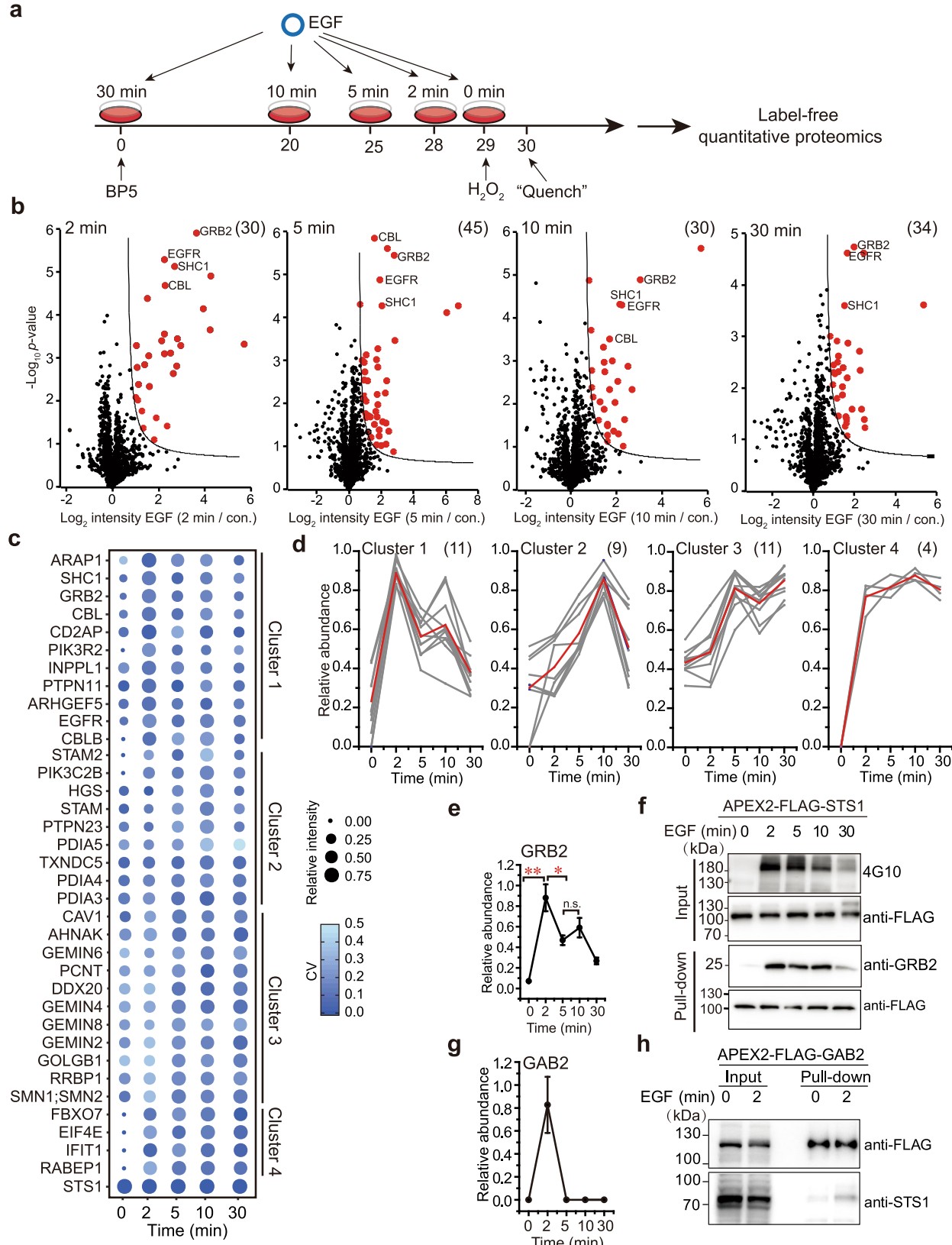

continuous increasing toward 30 min, with enriched GO annotation related to spliceosome; (4) interestingly, cluster 4, in which we observed 4 proteins, including RABEP1, IFIT1, EIF4E, and FBXO7, has EGF stimulation-dependent association with STS1 and reaches to plateau right after the stimulation. The curve for GRB2 in cluster 1 is accurate as indicated by the tight error bar

and independent validation by western blot (Fig. 4e, f). In summary, we built a temporal interactome map of STS1 in living cells by the selective BP5-based proximity proteomics approach.

Weak and transient pTyr-dependent signaling complexes often appear at the early stage of EGF activation, which is often missing without stimulation and therefore could not pass our stringent

**Fig. 4 Minute-resolution temporal profiling of the STS1 interactome. a** Workflow of proximity proteomic analysis with five EGF stimulation time-course. HeLa cell line with stably expressed APEX2-FLAG-STS1 were stimulated with EGF and labeled with BP5 as indicated. **b** Volcano plots of EGF stimulation-dependent STS1 interactome at each stimulation point. Significantly changed proteins are marked in red ($n = 3$ independent biological experiments). HeLa cells without EGF stimulation was used as common control. **c**. The temporal profiles of the STS1 interactome in living cells. The size of each dot is proportional to the relative abundance of the 35 interacting proteins. The CV (coefficient of variation) distribution is color-coded with different intensities. **d**. Four manually curated clusters of the STS1 temporal interactome according to curve similarity. The numbers and red curves indicate the proteins in each cluster and the average values of all the proteins, respectively. **e, f** Validation of STS1 interaction with GRB2 by streptavidin pull-down of APEX2-FLAG-STS1 and western blots ($n = 3$ independent biological experiments). Data are presented as mean values ± s.d. (error bars). *$p$-value is 0.0359. **$p$-value < 0.0001. n.s. not significant (according to two-sided Student's $t$ tests with adjustments made for multiple comparisons). **g, h** Validation of STS1 interacts with GAB2 by streptavidin pull-down of APEX2-FLAG-GAB2 and western blots ($n = 3$ independent biological experiments). Data are presented as mean values ± s.d. (error bars). Quantification of the WBs in **f** and **h** were presented in Supplementary Fig. 12b, c. Source data are provided as a Source Data file.

data analysis workflow (Supplementary Fig. 7c). In total, we curated 187 this type of proteins which had missing values at 0 min but could be well quantified at 2 min (Supplementary Fig. 8a). Among these proteins with diverse molecular functions and major localization at cytosol (Supplementary Fig. 8b, c), we were specifically interested in 26 signaling proteins annotated by the HPRD database, including a number of proteins with annotated association to EGFR signaling, such as adapter proteins (GAB1, GAB2, and CRK), kinases (AKT2), PI3K family members (PIK3CA) and GTPase regulators (SOS1, ARAP3, and ARHGEF12) (highlighted in Supplementary Fig. 8a). Among them, GAB2 is critical signaling protein for scaffolding membrane receptors and the intracellular signaling network through dynamic tyrosine phosphorylation[34]. Its weak and transient association with STS1 upon EGF stimulation was successfully captured by proximity proteomics and validated by reciprocal proximity labeling and western blot analysis (Fig. 4g, h).

**Comparison with AP-MS for temporal interactome profiling**. The use of the APEX-based proximity labeling approach for exploring temporal interactome in living cells is still in its infancy, with only two pioneering reports for profiling membrane-embedded GPCR complexes[15,16]. It is of interest to examine how well such data would match data obtained by AP-MS, which is the most popular approach for studying the dynamic cellular interactome. The BioID and AP-MS approaches to study stable interactome differ in principle, and comparison often shows only partial overlap[10]. We addressed this question by taking advantage of the APEX2 tag and FLAG tag in the same HeLa cell line stably expressed APEX2-FLAG-STS1 (Supplementary Fig. 9a).

Using the same data processing workflow (Supplementary Fig. 7c), FLAG tag-based AP-MS identified 14 STS1-interacting proteins, mostly with known functions in EGFR signaling, and quantified their temporal changes across 5 continuous time points of EGF stimulation (Fig. 5a, Supplementary Figs. 9b and 11j, and Supplementary Data 10). Surprisingly, the majority of the proteins identified by AP-MS could also be identified by the proximity proteomics with consistent time-course curves and close connections as annotated in the STRING database (Fig. 5b and Supplementary Fig. 9c). Although they did not meet the stringent cutoff in either proximity proteomics or AP-MS data, we manually included CSK, CD2AP, INPPL1, and CBLB in the overlap because of their similar curves in both methods. The two proteins unique to the AP-MS, ERBB2, and SH3KBP1, are known EGFR signaling proteins (Fig. 5c). ERBB2 was also identified in the APEX2-BP5 experiment but did not pass the stringent cutoff of our data analysis workflow. Interestingly, SH3KBP1 (CIN85) has the same tandem SH3 domain structure as CD2AP but shows completely different curves between the two methods, which might reflect the difference between interactions in vitro and in living cells.

We went on to select CD2AP which has distinct time-course curve comparing with its family member CIN85 and successfully validated its EGF stimulation-dependent interaction with STS1 by reciprocal proximity labeling and western blot analysis (Fig. 5d and Supplementary Fig. 9d). It should be noted that the stable interaction of CD2AP with STS1 has also been confirmed recently by the BioID approach in leukemia cells[35]. Furthermore, we validated colocalization of CD2AP with STS1 in EGF-induced membrane ruffles and the leading edges of cells (Fig. 5e). It should be highlighted that only a limited fraction of CD2AP and STS1 were recruited to plasma membrane upon EGF stimulation as indicated by the immunofluorescence, leaving majority of the proteins in cytosol. This result is also consistent with the EGF stimulation-dependent localization of GRB2 (Supplementary Fig. 3i) and should explain the identification of other cytosolic STS1 interactome beside its known association with membrane-associated EGFR signaling complexes. To further validate this conclusion, we stably expressed APEX2-FLAG-STS1 in HeLa cells with W295A mutation in its SH3 domain which is known to abolish its interaction with EGFR (Supplementary Fig. 9e)[31,36]. With its superior performance for high-selective labeling of cytosolic protein complexes over BP5, we adopted BN2-based proximity proteomic analysis and observed EGFR-independent association of STS1 and GRB2 in cytosol (Supplementary Fig. 9f and Supplementary Fig. 11k). In conclusion, proximity proteomics has great consistency with AP-MS in charting core EGFR signaling complexes but is much more comprehensive than AP-MS for profiling STS1 signaling protein complexes especially localized in other subcellular components.

**Spatiotemporal interactome profiling by BP5-based proximity proteomics**. We further explored the 24 dynamically regulated proteins that were captured only by the proximity proteomics approach. As shown in Fig. 6a, b, these proteins showed diverse subcellular localization, as annotated in the GO database or related references[37], and largely reached peak values at later time points of EGF stimulation (Supplementary Fig. 10). Representative subcellular locations include endosome, endoplasmic reticulum, and coiled bodies. Consistent with EGF-induced internalization of EGFR, endosome is the location containing the highest number of STS1-interacting proteins. Well-known endosome markers, including HGS, STAM and STAM2, were identified with well-behaved curves that reached a peak at 10 min of EGF stimulation (Fig. 6c). This result is consistent with recent reports in which density gradient centrifugation combined with MS analysis showed that STS1 and other EGFR-associated adapter proteins are enriched in endosomes after ~20 min of EGF stimulation in HeLa cells[37,38]. Among the endosome-enriched proteins, RABEP1 is a key factor in regulating endosome fusion and transportation[39,40]. Consistently, the time-course curve of RABEP1 in our study had peak values at later times of EGF stimulation, indicating its potential association in the STS1

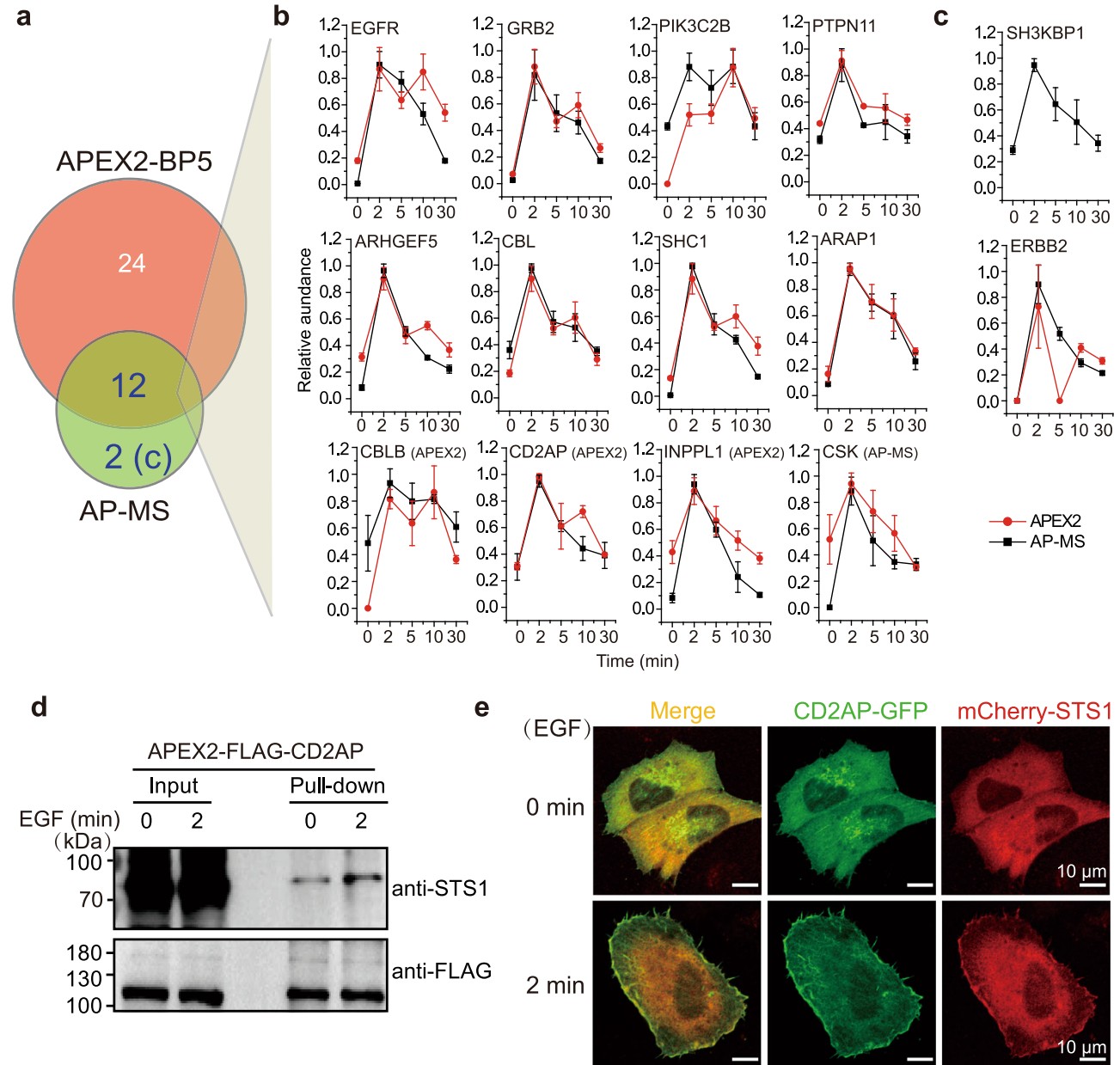

**Fig. 5 Comparison with AP-MS confirmed the dynamic and weak in vivo interactome of STS1. a** Comparison of temporal proteins identified by BP5-based proximity proteomics and FLAG tag-based AP-MS. AP-MS was performed in the same HeLa cells with stably expressed APEX-FLAG-STS1 and EGF stimulation as indicated in Fig. 4a. **b** Individual temporal curves of the STS1-interacting proteins identified in both APEX2-BP5 proximity proteomics and AP-MS. CBLB, CD2AP, INPPL1, and CSK meet the cutoff only in APEX2-BP5 proximity proteomics or AP-MS (as indicated in brackets), but were manually included into overlap part because of their curve similarity. **c** Individual temporal curves of the STS1-interacting proteins uniquely significant in AP-MS. Data in **b** and **c** are presented as mean values ± s.d. (error bars; n = 3 independent biological experiments). **d** Validation of CD2AP as an EGF-stimulation-dependent STS1-interacting protein by streptavidin pull-down of stably expressed APEX2-FLAG-CD2AP in HeLa cells and western blots (n = 3 independent biological experiments). Quantification was presented in Supplementary Fig. 12d. **e** Colocalization of transiently co-transfected GFP-tagged CD2AP and mCherry-tagged STS1 in HeLa cells upon EGF stimulation (n = 3 independent experiments). The scale bars are 10 μm. Results were quantitatively confirmed by line profile analysis shown in Supplementary Fig. 12b. Source data are provided as a Source Data file.

complex for subcellular trafficking. Immunofluorescence analysis showed typical localization of RABEP1 in subcellular structures and partial colocalization with STS1 at 10 min of EGF stimulation (Fig. 6d).

## Discussion

In this study, we developed a highly selective APEX2-based proximity proteomics approach for charting the spatiotemporal interactome with minute timescale resolution. Our developed biotin-phenol analog probes BP5 and BN2 generates free radicals and conjugates to tyrosine residues in proteins more efficiently and selectively than the previously reported BP1. For BP5, this is largely due to the moderate electron donating activity of the para-amide group in BP5, which facilitates electron transfer to the phenol hydroxyl group and therefore O–H bond dissociation[19,20]. Its higher labeling selectivity than BP1 is reasonable considering to its lower BDE for generating radical and higher reactivity for

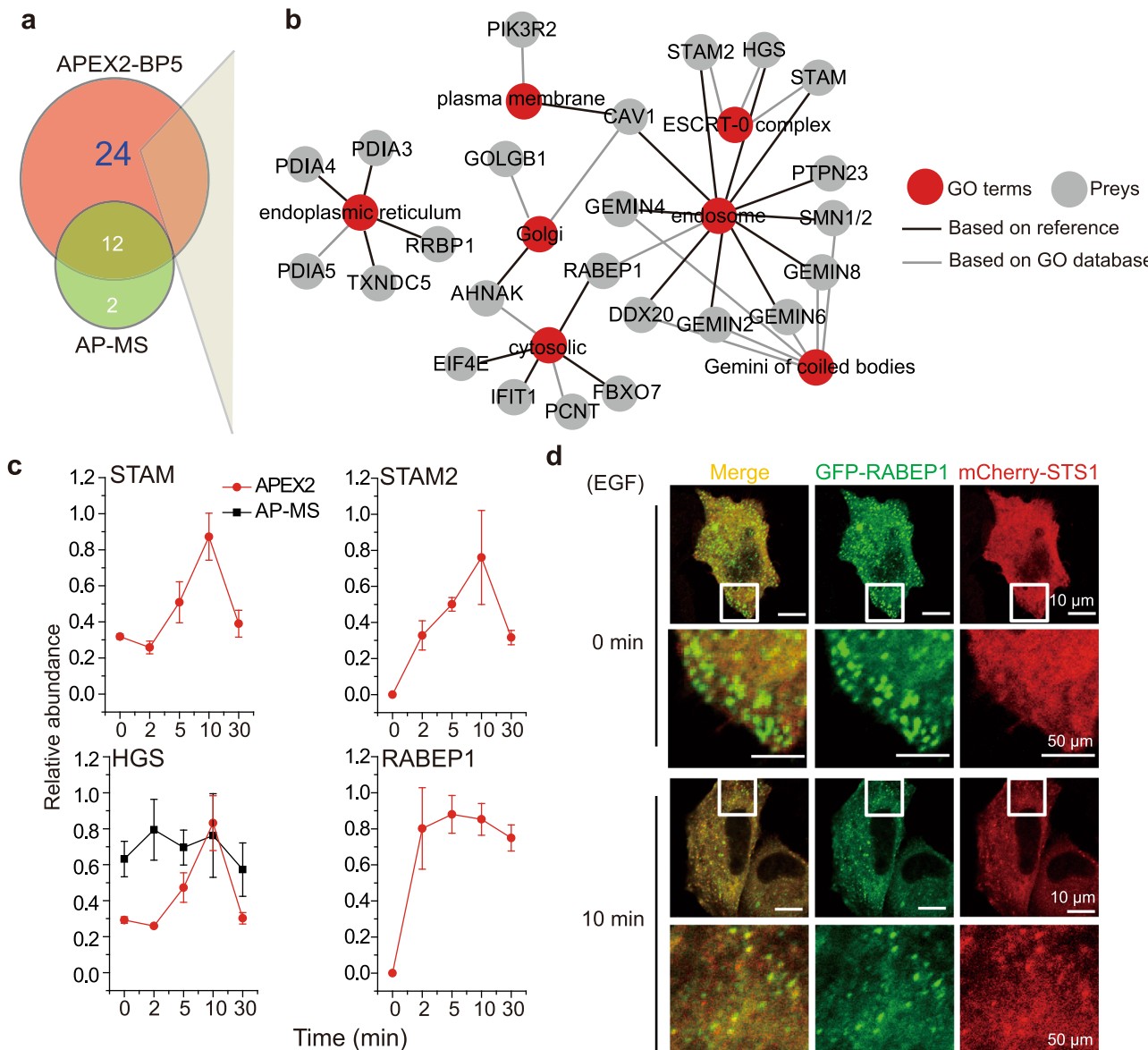

**Fig. 6 Spatiotemporal STS1 interactome at late time points of EGF stimulation. a, b** Overview and subcellular localization of STS1-interacting proteins that pass the data analysis cutoff only in APEX2-BP5 proximity proteomics. **c** The temporal curves of four proteins which are pivotal for membrane protein trafficking and endosome targeting. Data are presented as mean values ± s.d. (error bars; $n = 3$ independent biological experiments). **d** Colocalization of transiently co-transfected GFP-tagged RABEP1 with mCherry-tagged STS1 in HeLa cells upon EGF stimulation. HeLa cells were co-transfected with GFP-tagged RABEP1 and mCherry-tagged STS1 ($n = 3$ independent biological experiments). The scale bars are 10 μm and 50 μm (for zooming in). Results were quantitatively confirmed by line profile analysis shown in Supplementary Fig. 13c. Source data are provided as a Source Data file.

forming BP5 trimer which prohibit it to diffuse farther out and label background noise. BN2 has been shown to have strong APEX2-mediated labeling activity for DNA but weaker labeling activity than biotin-phenol for proteins[18]. Side-by-side comparison between BN2, BP5, and BP1 in this study showed that BN2 could label cytosolic IPP complex with much higher selectivity which is largely due to its higher reactivity and robust formation of polymer which is in agree with the recent report by Liu et al.[21] In summary, BP5 and BN2 should be used in a complimentary manner for labeling cytosolic protein complexes within different radius from the background noise.

Temporal interactome profiling with minute resolution in living cells has been conducted for membrane-associated GPCRs by adopting BP1, APEX2-based proximity labeling and quantitative proteomics[15,16]. By setting known "bystanders" with specific

locations as controls, the spatiotemporal interactome of GPCRs was well characterized. However, as highlighted by Lobingier et al.[16], such an approach is still not feasible for studying the spatiotemporal interactome of cytoplasmic proteins that lack specific organelle locations. In this study, by taking advantage of the BP5- and BN2-based proximity labeling with high selectivity, we successfully applied the proximity proteomics approach to explore dynamic protein complexes associated with 5 cytosolic proteins with partly association with membrane-embedded EGFR, including GRB2, STS1, SHC1, CD2AP, and GAB2, and the spatiotemporal interactome of STS1. In addition, we also explored the protein complexes associated with four cytosolic proteins, including ILK, RSU1, PINCH, and PARVIN. The stable interactome of STS1 has been reported by us and others recently based on both AP-MS[41,42] and the BioID approach[35]. In this

study, we systematically extended the exploration of the STS1 interactome in a spatiotemporal manner with minute resolution.

AP-MS is still the most popular approach for exploring the temporal interactome of specific proteins, especially cytosolic proteins. Our proximity proteomics methods provides a useful alternative when the interactome profiling in living cells with spatiotemporal and minute resolution is of interest. We provide a side-by-side comparison between the AP-MS and proximity proteomics approaches with multiple time points and high quantification precision. AP-MS is reasonably good for profiling core EGFR signaling complexes. In comparison, our proximity proteomics approach captured many more interacting proteins especially with distinct spatial localization and potential weak association. However, our proximity proteomics approach still has limitations. First, by modulating the BDE of the O–H on the phenol of biotin-phenol, we observed only a 2–3-fold increase in labeling selectivity for charting dynamic interactome compared with that of traditional biotin-phenol. The complimentary application of BN2 could significantly improve the labeling selectivity when cytosolic protein complex with direct association and in smaller radius is of interest. Engineering APEX2 or exploring new peroxidases for tuning the labeling activity toward protein complex profiling applications might further improve the labeling selectivity as well. Second, our proximity proteomics approach still relies on activation by 500 μM $H_2O_2$, which might limit its application for exploring other spatiotemporal interactomes in living cells which are sensitive to $H_2O_2$ treatment. Further improvement of the proximity labeling mechanism by skipping $H_2O_2$ activation, as Branon et al.[12] did for TurboID, would be highly promising. In conclusion, our proximity proteomics approach provides a complementary approach for exploring the spatiotemporal interactome in living cells when minute resolution is needed, such as in pTyr-mediated signaling networks.

## Methods

**Synthesis and screening of biotin-phenol derivatives**. The detailed synthesis procedures of biotin-phenol derivatives were described in the Supplementary Methods. The in vitro screening of biotin-phenol derivatives was performed as reported with minor revision[13]: 500 μM of different probes, 10 nM HRP (SIGMA, P8375), and 1 mM $H_2O_2$ in Dulbecco's Phosphate-Buffered Saline (DPBS, pH 7.4) were mixed together. The reaction was triggered by $H_2O_2$ in experimental groups for 1 min or 20 min at 37 °C, and then quenched by 1 mM sodium azide. For comparison between BP1 and BP5 in presence of tyrosine, reactions were assembled as follows: 500 μM BP1 and/or 500 μM BP5, 500 μM tyrosine and 10 nM HRP were mixed in DPBS. The reaction was triggered by adding 1 mM $H_2O_2$ for 1 min or 20 min at 37 °C, and quenched by 1 mM sodium azide. BP5 or BN2 reaction with synthetic FLAG peptide (MDYKDDDDK) and CD28 tryptic peptide (KHYQPYAPPR) with or without phosphorylation at the second tyrosine were performed with the same reaction design as the reaction with tyrosine. Product mixtures were analyzed by LCQ-Fleet equipped with Ultimate 3000 HPLC system (Thermo Fisher Scientific). The HPLC column used in this study was Hypersil GOLD C18 column (1.9 μm, 2.1 × 100 mm, Thermo Fisher Scientific) with a gradient of 0–90% acetonitrile in 0.1% (v/v) formic acid /water for 23 min. The BP analogs and their products were identified by MS and quantified by chromatography peak area. The identity of compounds was confirmed by Q-Exactive Orbitrap mass spectrometer with mass resolution of 70,000 (Thermo Fisher Scientific).

**Plasmids and cell lines**. We adopted the lentivirus system with characteristics of doxycycline-inducible expression and puromycin resistance for stable cell line construction[43]. To ensure a flexible environment for bait protein folding, we transformed the original expressing plasmids with a FLAG tag between the bait protein and the APEX2, leaving a PacI/PmeI subcloning site C-terminal to the APEX2-FLAG sequence. The coding sequence of GRB2 (CR_541942.1), ILK (NM_001014794.3), RSU1 (NM_012425.4), PINCH (NM_001193485.3), SHC1 (NM_183001.4), STS1 (NM_032873.4), CD2AP (NM_012120.2), GAB2 (BC131711), RABEP1 (NM_007403.5) and GFP (AUM57423.1) were cloned into the expressing plasmid with PacI/PmeI restriction sites. W295A mutant of STS1 was cloned into the lentivirus plasmid by converting the original base sequence TGG (code for W) to GCG (code for A). The recombinant plasmids and the virus packaging plasmids (pRSV/REV, pMDLg/pRRE and pCMV-VSVG) were co-

transfected into 293T cell to produce virus. After incubation for 72 h, the virus was harvested for HeLa cells or HT1080 cells (ILK, RSU1 and PINCH) infection for 6 h. The 293 T, HeLa and HT1080 cell lines were purchased from American Type Culture Collection (ATCC). After infection, the culture medium was replaced with fresh medium containing 10% (v/v) FBS and 1% (v/v) penicillin-streptomycin solution (Corning, 30-002-Cl).

For immunofluorescence, GRB2 and RABEP1 were cloned into pEGFP-C2 vector (Clontech) using EcoRI and SalI restriction sites. CD2AP was cloned into pEGFP-N1 vector (Clontech) using EcoRI and SalI restriction sites. PCR-amplified mCherry (AIJ27453.1) was cloned into the pcDNA 3.1 myc-his B vector (Thermo Fisher Scientific) with BamHI restriction site, then STS1 was cloned into the modified plasmid containing an N-terminal mCherry tag using EcoR1 and XhoI restriction sites. All the primers for cloning were listed in Supplementary Data 11.

**Cell treatment and proximity labeling**. HeLa cell lines stably expressing APEX2-FLAG tagged proteins were cultured in DMEM containing 10% (v/v) FBS and 0.6 μg/mL puromycine (Thermo Fisher Scientific, A1113803). HT1080 stable cell lines were cultured in MEM containing 10% (v/v) FBS and 1 μg/mL puromycine. After 24 h of induction for protein expression with 1 μg/mL Dox, 500 μM BP analogs were incubated with the cells for 30 min before further treatment. The HeLa cells were starved in DMEM alone for 4 h before incubation with 500 μM probes for 29 min. For 2 min EGF stimulation, 100 ng/mL EGF was added after 28 min of probe incubation. Proximity labeling was triggered precisely for 1 min by the addition of 500 μM $H_2O_2$ when the 1 min of probe incubation time remained. The labeling was quenched by ice-cold quenching buffer (10 mM sodium azide, 10 mM sodium ascorbate, 5 mM trolox, dissolved in DPBS), and then washed three times with ice-cold DPBS. In proximity labeling experiment, the cells were lysed with 700 μL of ice-cold RIPA lysis buffer [50 mM Tris-HCl (pH 8.0), 150 mM NaCl, 0.1% (w/v) SDS, 0.5% (w/v) sodium deoxycholate, 1% (v/v) Triton X-100, 1 mM EDTA, 10 mM sodium azide, and 10 mM sodium ascorbate, 5 mM Trolox, 1 mM $Na_3VO_4$ and 50 mM PMSF], sonicated, and then centrifuged at 16,612 × g and 4 °C for 10 min. In FLAG IP experiment, cells were stimulated by EGF at the same time points as described in proximity labeling, and the cell was lysed in mild lysis buffer [50 mM Tris-HCl (pH 7.2), 150 mM NaCl, 1% (v/v) Triton X-100, 1 mM EDTA, 10 mM sodium azide, and 1 mM $Na_3VO_4$ and 50 mM PMSF] with the same volume.

**Western blot and immunofluorescence analysis**. Equal amounts of samples were loaded for western blot and specific proteins were detected with corresponding antibodies. The primary antibodies used in this study were 4G10 (Merck Millipore, 05-321, 1:1000), anti-pEGFR (Tyr1068; CST, 3777 s, 1:1000), anti-pERK1/2 (CST, 9101, 1:1000), anti-ERK1/2 (CST, 4695, 1:1000), streptavidin-HRP (Thermo Fisher Scientific, 21130, 1:3000), anti-FLAG (SIGMA, F1804, 1:1000), anti-EGFR (CST, 4267 s, 1:1000), anti-CBL (CST, 8447 s, 1:1000), anti-GRB2 (BD, 610112, 1:1000), anti-STS1 (Abcam, ab34781, 1:1000), anti-SHC (BD, 610878, 1:1000), and anti-β-actin (Beyotime, AF0003, 1:1000). The secondary antibodies include HRP-conjugated anti-rabbit (Beyotime, A0208, 1:1000) and anti-mouse (Beyotime, A0216, 1:1000). All WB images were collected using Tanon 6100 C (SN 14T15RGBFLI6-1226) image system. All WBs were performed with at least three independent experiments, and quantified with ImageJ software (version ij153-win-java8). Uncropped gel images and replicates were included in the Source Data file, and the quantification were presented in Supplementary Fig. 12.

For immunofluorescence analysis, HeLa cells were plated on glass coverslips (Fisherbrand, LOT17951) and DMEM supplemented with 10% (v/v) FBS for 24 h at 37 °C and 5% $CO_2$. Then, the cells were transfected with 2 μg plasmids for 24 hours using Lipofectamine[3000] and starved by serum-free medium for 4 h before stimulated with 100 ng/mL EGF for 2 min (for GRB2, STS1 and CD2AP experiments) or 10 min (for RABEP1 experiments); the cells were then washed twice with DPBS and fixed with 4% (v/v) paraformaldehyde for 10 min. Protein localization was detected by GFP and mCherry fluorescence using an inverted fluorescence microscope (Nikon-TiE) equipped with 10 × 0.45 NA dry and 100 × 1.4 NA oil immersion objective lens, 2 laser lines (488 nm and 561 nm). All images were analyzed using Nikon Nis-element AR (version 4.40.00) and ImageJ software. All immunofluorescence were performed with three independent experiments, and the results were quantitatively confirmed by line profile analysis using ImageJ software (version ij153-win-java8) (Supplementary Fig. 13 and Source Data file).

**Pull-down and MS analysis**. Pull-down was performed by incubating 2.5 mg of cell lysate with 60 μL of streptavidin sepharose (GE, 17-5113-01) in proximity labeling experiment or 30 μL of anti-FLAG M2 affinity gel (Merck Millipore, A2220) in AP-MS experiment. After incubation overnight at 4 °C, streptavidin beads were then washed with the following procedures: twice with RIPA lysis buffer, once with 2 M urea in 10 mM Tris-HCl, pH 8.0, and two more times with RIPA lysis buffer. The anti-FLAG affinity gel was washed with mild lysis buffer for three times. The beads were then washed three times with 25 mM ammonium bicarbonate. On-bead digestion and label-free quantitative proteomics analysis were performed as we described previously with more details in the Supplementary Methods[6]. Briefly, on-bead digestion was performed by adding dithiothreitol, iodoacetamide and trypsin (1.5 μg, Promega, V5111). Digested peptides were desalted and redissolved for nano LC-MS/MS analysis on an Orbitrap Fusion mass

spectrometer or Q Exactive HF-X (for comparison between BN2 and BP5) equipped with Easy-nanoLC (Thermo Fisher Scientific). All the MS data were collected using Thermo Scientific Xcalibur software (version 4.1.50). More details about MS sample preparation, biotinylated peptide enrichment, and MS analysis are included in the Supplementary Methods.

**Data analysis.** Database search and label-free quantification by MaxQuant software (version 1.5.5.1) and downstream statistical analysis by Perseus software (version 1.5.5.3) were done as our previous report and as described in the Supplementary Methods[6]. Proteins identified with at least two unique peptides were taken into consideration for further data analysis. All the volcano plots were created in the Perseus software and evaluated by two-sided Student's $t$ test with a permutation-based FDR value of 0.05 and S0 value of 0.5[44]. In the STS1 interactome time-course study, the proteins with different characteristics were manually clustered based on their curve similarity. The interacting protein curves were constructed after normalization based on the LFQ intensity of the bait protein STS1. Minimum of 2 valid values (after $\log_2$ transformation) was kept for CV calculation, and the LFQ intensity of proteins with only one valid value was set to zero. The protein-protein relationship was analyzed with STRING (version 11.0; https://string-db.org/). Molecular function and cellular compartment annotation were referred to GO knowledgebase (Released at 2020-06-01, with 44,411 GO terms and 7,975,639 annotations; FDR < 0.05; http://geneontology.org/)[45]. The unannotated proteins in Fig. 6b were annotated according to a related report[37]. More details about the data analysis are described in the Supplementary Methods.

**Reporting summary.** Further information on research design is available in the Nature Research Reporting Summary linked to this article.

## Data availability

All the raw MS data have been deposited to ProteomeXchange Consortium repository[46] with the dataset identifier PXD020709. Source data are provided with this paper.

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

## Acknowledgements

We are grateful to Chuan He (University of Chicago) and Benjamin Neel (NYU Langone Health) for providing comments on manuscript. The authors acknowledge funding from

National Natural Science Foundation of China (91953118), China State Key Basic Research Program Grants (2016YFA0501403 and 2016YFA0501404), Guangdong Provincial Fund for Distinguished Young Scholars (2019B151502050), Guangdong Provincial Natural Science Grant (2016A030312016), Shenzhen Innovation of Science and Technology Commission (JCYJ20170412154126026).

## Author contributions

M.K., X.Y. and R.T. designed the experiments. M.K. performed most of the experiments. X.Y. performed in vitro and immunofluorescence experiments. A.H. designed and synthesized the probes. A.H., M.K. and X.Y. analyzed the data. P.Y. performed the DFT computations. W.C. helped with biotinylated peptide enrichment experiment. Y.S., T.H. and P.Z. provided reagents and suggestions. R.T. conceived and supervised the project. M.K., X.Y. and R.T. wrote the paper.

## Competing interests

The authors declare no competing interests.

## Additional information

**Peer review information**:*Nature Communications* thanks the anonymous reviewers for their contribution to the peer review of this work. Peer reviewer reports are available.

