## [Peer Review File · Nature Communications]

REVIEWER COMMENTS

Reviewer #1 (Remarks to the Author):

In the submitted manuscript, the authors has profiled GRB2 and STS1 interactome by APEX2 with new chemical probes. It is impressive that the authors monitored minute-resolution temporal interactome of STS1 by EGF stimulation. Since current manuscript is full of high quality data, this reviewer expect that this manuscript might be suitable to be published in Nature Communications after several things in the current manuscript is further clarified.

1. To support the experimental observations of spatial specificity of BP5 radical over BP1 radical, the authors have provided DFT-calculated values (i.e. BDE and energy level of transition state) of the coupling reaction of each radical probes on the tyrosine. However, it is required to add enough explanation how these values could be linked to the lifetimes or spatial specificity of the radical probes.

2. In main text, the authors mentioned that it is possible to identify hundreds of BP5-modified peptides with APEX2-GRB2 (Supp Fig 1L) and also provided self-labeled peptide sequence of APEX2-GRB2 (Supp Figure 1m). This reviewer believe that the data of BP5-modified peptidome should tell the most correct labeled proteome information of APEX2-GRB2, however, this BP5-modified peptidome information is not available as a separated datasheet file in current submission. The data of the BP5-modified peptidome should be uploaded as a separate Supplementary dataset and its processing workflow from the raw data should be described in Supplementary Information. and it would be also great if the authors can comment whether the BP5-modified peptidome information of APEX2-GRB2 is well-overlapped with the analysed data from the unlabeled peptidome detection (Fig 2b).

3. It is impressive that the authors have profiled temporal interactome of STS1 after EGF stimulation and confirmed their findings by imaging experiments: CD2AP-GFP in Figure 5e, GFP-RABEP1 in Figure 6d with mCherry-STS1. Although the authors already showed that CD2AP and RABEP1 are colocalized with mCherry-STS1 at the different time points in "different" cells after EGF stimulation,

it might be more impressive if the authors can conduct the live cell imaging experiments and provide real-time imaging movies that CD2AP-GFP and GFP-RABEP1 are co-localized with mCherry-STS1 in the "same" cells after stimulation of EGF, respectively.

Reviewer #2 (Remarks to the Author):

In the revised version of their manuscript, the authors have gone a long way to address the criticism and concerns that I had. To do this, the authors have generated important new datasets that significantly improves the manuscript. For example, they substantially expanded their analysis to include more APEX substrates (12) including five novel ones. They screened these in a proximity labeling setting and included a much more extensive proximity labeling MS analysis of the most promising substrates, BP5 and BN2, showing that they could serve as complimentary approaches to reveal relatively distant versus closer proximity proteins in complexes, respectively. In particular, the new BN2 analog looks very promising with its self-quenching properties and higher selectivity for direct interactors, it looks like it is the one that can actually be used to capture cytosolic complexes such as the ILK-PINCH-RSU1 complex presented (and not only membrane-bound complexes like for BP1 and BP5). I would therefore recommend the authors to rewrite the main manuscript text and change the focus from the biological findings (i.e. description of the STS1 interaction), and instead shift emphasis to the methods-oriented improvements and in particular make the comparisons of BP1, BP5 and BN2, the red line throughout the manuscript. Such a change to a more method-based focus of the manuscript would compensate for the limited novel biological insights, and instead highlight the important new finding that BN2 captures closer proximity proteins compared to BP5, which is interesting and of high value for the proximity ligation community.

Other (minor) comments that should be addressed before I can recommend publication:

- The statement about STS1 as previously underappreciated – no evidence for the opposite, biological meaning missing. Suggest toning down this part of the manuscript as it mainly serves as validation of already known biology.
- Tables with information on all identified proteins in BN2 versus BP5 Grb2-EGF proximity labeling experiment is missing. The identity of these proteins that according to the authors serve as background/noise level as the authors state are important to actually deem them as background noise.
- Have the authors performed any comparison of differences in the noise level between the different substrates?

- The authors claim that the key mechanism of BP5 and BN2 is to reduce the background protein labelling to increase signal to noise. However, they never perform an analysis to reveal the identity of these background binders and compare their intensity between BP1, BP5 and BN2. Such an analysis should be performed to understand why BP5 and BN2 captures more significant interactors than BP1.
- A change to density-coloring in the volcano plots would give a more fair representation of the high number of background binders. For example, in Fig. S10 it is evident that thousands of proteins are identified in each proximity labeling analysis. However, this is not evident from any of the volcano plots – a density coloring should serve as a proper visualization.
- The authors still did not provide quantitation of WBs, although this was stressed in the original review. Quantification of all WB with n=3 should be performed. Importantly Fig 1d and Fig S3c!
- The schematic Fig 1b should clearly state what is in-vitro and what is in-vivo – currently this is indistinguishable.
- The schematic representations (Fig. 2a and 3a) should include a description of the APEX part and the dots (biotinylation) to improve the readability.
- Fig. 4e – is this levels of multiple significance testing necessary?
- IF co-localization should be quantified Fig 5e, Fig 6d
- p.5 line 104: The authors perform an analysis of probe dimer and trimer formation – it is here to clear to the reader why. Please provide a sentence why this is important (eg. p. 7 line 138)
- Several figures and figure legends still need an indication of whether they were performed as min. three independent experiments. Especially in the supplementary figures.

Point-by-point responses to individual comments:**Reviewer #1 (Remarks to the Author):**

In the submitted manuscript, the authors has profiled GRB2 and STS1 interactome by APEX2 with new chemical probes. It is impressive that the authors monitored minute-resolution temporal interactome of STS1 by EGF stimulation. Since current manuscript is full of high quality data, this reviewer expect that this manuscript might be suitable to be published in Nature Communications after several things in the current manuscript is further clarified.

1. To support the experimental observations of spatial specificity of BP5 radical over BP1 radical, the authors have provided DFT-calculated values (i.e. BDE and energy level of transition state) of the coupling reaction of each radical probes on the tyrosine. However, it is required to add enough explanation how these values could be linked to the lifetimes or spatial specificity of the radical probes.

We thank the reviewer's encouraging comments and are sorry for our inadequate discussion about density functional theory (DFT) computations largely due to the limited manuscript space. To our current understanding, the proximity labeling could be attributed by two reaction steps: (1) radical generation as determined by BDE and (2) radical reaction between probe and tyrosine residue as determined by ΔG^\ddagger . Therefore, proximity labeling lifetime and selectivity are determined by the first and second steps of reaction in a combinatory manner.

Specifically, lower BDE is favorite for more efficient radical generation, while lower ΔG^\ddagger renders shorter half-life and labeling radius due to faster radical reaction and self-quenching before the radical could diffuse farther out. In the DFT calculation, BP5 showed the lowest BDE and relatively lower ΔG^\ddagger among tyrosine, BP1, BP4, BP5, BN1 and BN2, indicating the lowest barrier for generating radical and shorter labeling radius due to the efficient formation of BP5 dimer or trimer. Similarly, although BN2 has higher BDE than BP1, the lowest ΔG^\ddagger of BN2 for reacting with phenol radical and robust formation of polymer ensure it shorter labeling radius compared to BP1. Our experimental results in **Figure 1-3** were largely consistent with these theoretical DFT computations. Accordingly, we have improved the related discussion and figure legend description in the revised manuscript.

2. In main text, the authors mentioned that it is possible to identify hundreds of BP5-modified peptides with APEX2-GRB2 (Supp Fig 1L) and also provided self-labeled peptide sequence of APEX2-GRB2 (Supp Figure 1m). This reviewer believe that the data of BP5-modified peptidome should tell the most correct labeled proteome information of APEX2-GRB2, however, this BP5-modified peptidome information is not available as a separated datasheet file in current submission. The data of the BP5-modified peptidome should be uploaded as a separate Supplementary

dataset and its processing workflow from the raw data should be described in Supplementary Information. and it would be also great if the authors can comment whether the BP5-modified peptidome information of APEX2-GRB2 is well-overlapped with the analyzed data from the unlabeled peptidome detection (Fig 2b).

We thank the reviewer's kind comment for improving this supplementary figure. Accordingly, we have now uploaded the BP5-modified peptidome data as **Supplementary data 4** and added related experimental description in the revised manuscript. Specifically, 650 BP5-modified peptides and 912 PSMs were identified, which belong to 435 proteins. In comparison with the proximity proteomics data for APEX2-GRB2 in **Fig. 2b**, GRB2 and 11 interacting proteins were identified with BP5-modified peptides, including STS1, CD2AP, CRK, WIPF2, EZR, WASL, EIF3J, CBL, PTPN11, ANKS1A and ZDHHC5. Although the BP5-modified peptidome analysis successfully identified a fraction of GRB2 interactome, the interactome and protein sequence coverage are limited due to the low efficiency for enriching modified peptide directly from digested cell lysate. As we presented in **Supplementary Figure 4a**, the total identification number of biotinylated peptides and enrichment efficiency have been optimized to the well-accepted level. The introduction of new enrichment strategy is therefore expected for better application of biotinylated peptidome analysis in protein complex study.

3. It is impressive that the authors have profiled temporal interactome of STS1 after EGF stimulation and confirmed their findings by imaging experiments: CD2AP-GFP in Figure 5e, GFP-RABEP1 in Figure 6d with mCherry-STS1. Although the authors already showed that CD2AP and RABEP1 are colocalized with mCherry-STS1 at the different time points in "different" cells after EGF stimulation, it might be more impressive if the authors can conduct the live cell imaging experiments and provide real-time imaging movies that CD2AP-GFP and GFP-RABEP1 are co-localized with mCherry-STS1 in the "same" cells after stimulation of EGF, respectively.

As agreed by the reviewer, our imaging data in **Fig. 5e** and **Fig. 6d** well validated the co-localization of CD2AP-STS1 and RABEP1-STS1 after EGF stimulation. Although different cells were analyzed at different time points of EGF stimulation, the fixed cell-based immunofluorescence imaging is well-accepted and the results from multiple cells should avoid possible artifact. We agree that real-time imaging of the same cell across different time points of EGF stimulation is ideal but technically challenging due to the efficient EGF stimulation of living cells on glass bottom cell culture dish and re-focusing of the high-resolution microscopy for dual-color monitoring afterward. Nevertheless, after several failure at our imaging facility, we finally obtained the live-cell imaging data by using the imaging facility of USTC at Hefei city. As shown in **Rebuttal Fig. 1**, CD2AP-GFP and mCherry-STS1 were diffusely distributed in the cytosol before EGF stimulation. When EGF was added to the cell, we successfully monitored the recruitment of CD2AP-GFP and mCherry-STS1 to the plasma membrane at 2 min and progressive declining from

2 min to 15 min. This live-cell imaging data in the “same” cell is consistent with the immunofluorescence imaging data collected in "different" cells (**Fig. 5e**). Unfortunately, we failed to capture the real-time imaging from 0 min to 2 min due to the re-focusing operation for the microscopy after adding EGF. In addition, we observed partial recruitment of CD2AP-GFP and mCherry-STS1 due to the less efficient EGF stimulation with special reagent deliver setup in a short operation time. With these technical concerns and requested focus of this study for method development, we decided to only present the real-time imaging data for CD2AP-STS1 co-localization in the rebuttal letter. For review purpose, we have also attached time-lapse movie of 14 consecutive time points after EGF stimulation from 2 min to 15 min.

Reviewer #2 (Remarks to the Author):

In the revised version of their manuscript, the authors have gone a long way to address the criticism and concerns that I had. To do this, the authors have generated important new datasets that significantly improves the manuscript. For example, they substantially expanded their analysis to include more APEX substrates (12) including five novel ones. They screened these in a proximity labeling setting and included a much more extensive proximity labeling MS analysis of the most promising substrates, BP5 and BN2, showing that they could serve as complimentary approaches to reveal relatively distant versus closer proximity proteins in complexes, respectively. In particular, the new BN2 analog looks very promising with its self-quenching properties and higher selectivity for direct interactors, it looks like it is the one that can actually be used to capture cytosolic complexes such as the ILK- PINCH-RSU1 complex presented (and not only membrane-bound complexes like for BP1 and BP5).

I would therefore recommend the authors to rewrite the main manuscript text and change the focus from the biological findings (i.e. description of the STS1 interaction), and instead shift emphasis to the methods-oriented improvements and in particular make the comparisons of BP1, BP5 and BN2, the red line throughout the manuscript. Such a change to a more method-based focus of the manuscript would compensate for the limited novel biological insights, and instead highlight the important new finding that BN2 captures closer proximity proteins compared to BP5, which is interesting and of high value for the proximity ligation community.

Other (minor) comments that should be addressed before I can recommend publication:

- The statement about STS1 as previously underappreciated – no evidence for the opposite, biological meaning missing. Suggest toning down this part of the manuscript as it mainly serves as validation of already known biology.

We appreciate the reviewer's positive comments to our revision and fully agree that the focus of this study should be changed to method development. Accordingly, we have toned down related statement for the novel discovery

of STS1 interactome throughout the whole manuscript. Instead, we now only treated STS1 as a known component of EGFR signaling complexes and used it as an example for demonstrating the BP5-based proximity labeling performance in time-course study and in comparison with AP-MS, especially for exploring spatial interactome. Related changes have been made in the revised manuscript.

- Tables with information on all identified proteins in BN2 versus BP5 GRB2-EGF proximity labeling experiment is missing. The identity of these proteins that according to the authors serve as background/noise level as the authors state are important to actually deem them as background noise.

We apologize for providing the requested dataset in the “Source Data” named “Supplementary Fig. 4g-h” in our last submission and have now moved the table to **Supplementary data 5**.

- Have the authors performed any comparison of differences in the noise level between the different substrates?
- The authors claim that the key mechanism of BP5 and BN2 is to reduce the background protein labelling to increase signal to noise. However, they never perform an analysis to reveal the identity of these background binders and compare their intensity between BP1, BP5 and BN2. Such an analysis should be performed to understand why BP5 and BN2 captures more significant interactors than BP1.

We thank the reviewer for this insightful suggestion for comparing the noise level across different probes. In fact, we have performed such type of comparison between BP1 and BP5 in **Fig. 2e**. By comparing the LFQ intensity of both probes for labeling all the cytosolic proteins and reported interacting proteins, we concluded that BP5 had higher selectivity for labeling real interacting proteins rather than non-relevant cytosolic proteins.

Upon the reviewer’s request, we now made further comparison for the background labeling of BP1 vs. BN2 and BP5 vs. BN2. As shown in the new **Supplementary Fig. 6e**, we systematically compared the background labeling as represented by APEX2-tagged GFP and concluded that BN2 outperform BP1 and BP5 for having significantly less background labeling.

- A change to density-coloring in the volcano plots would give a more fair representation of the high number of background binders. For example, in Fig. S10 it is evident that thousands of proteins are identified in each proximity labeling analysis. However, this is not evident from any of the volcano plots – a density coloring should serve as a proper visualization.

We thank for the reviewer’s comment for improving our data presentation in volcano plots. We agree that density-colored plot could better present the distribution of background noise for comparing different probes as presented

in **Supplementary Fig. 6e**. However, it cannot give much difference as compared with our current presentation when the same probe is applied for differentiating the interactome change before and after temporal EGF treatment. Accordingly, we made density-coloring volcano plots for all of our volcano plots and presented in **Rebuttal Fig. 2**. We got expected result as we discussed above for exploring temporal interactome with the same probe. As expected, we did observe different distribution of background noise when GFP was used as the control group. Since the density-coloring volcano plots could provide limited new information as compared with our current presentation, rather than complicate the presentation of significantly changed proteins, we decide to keep our current presentation.

- The authors still did not provide quantitation of WBs, although this was stressed in the original review. Quantification of all WB with n=3 should be performed. Importantly Fig 1d and Fig S3c!

We appreciate the reviewer to further stress the importance of WB quantitation and have now performed quantification analysis for all our WBs. Due to the limited space in our current figures, all the quantification data has been assembled in the new **Supplementary Fig. 12**. Related description has been added into the corresponding figure legend in the revised manuscript. All the raw data for biological triplicates has been uploaded in the "Source Data".

- The schematic Fig 1b should clearly state what is in-vitro and what is in-vivo – currently this is indistinguishable.

We apologize for the inadequate presentation and have removed the plasma membrane presentation for avoiding misunderstanding.

- The schematic representations (Fig. 2a and 3a) should include a description of the APEX part and the dots (biotinylation) to improve the readability.

We apologize for the inadequate description and have made related improvement in the revision.

- Fig. 4e – is this levels of multiple significance testing necessary?

We have removed redundant annotation in the revision.

- IF co-localization should be quantified Fig 5e, Fig 6d

We have now included quantification data for all of our imaging data related to **Fig. 5e**, **Fig. 6d** and **Supplementary Fig. 3i**. Due to the limited space in our current figures, we assembled the quantification data in the new

Supplementary Fig. 13 and add related annotation in the corresponding figure legends.

- p.5 line 104: The authors perform an analysis of probe dimer and trimer formation – it is here to clear to the reader why. Please provide a sentence why this is important (eg. p. 7 line 138)

We apologize for the inadequate description and have made related improvement in the revision.

- Several figures and figure legends still need an indication of whether they were performed as min. three independent experiments. Especially in the supplementary figures.

All the data was collected with minimal three independent experiments. We apologize for the inadequate description and added required information accordingly.

Rebuttal Figure 1. Live-cell fluorescence imaging of CD2AP-GFP and mCherry-STS1 in HeLa cell. The HeLa cells were cultured in 35 mm glass bottom cell culture dish (Corning), and then transfected with CD2AP-GFP and mCherry-STS1 plasmids. After 36 h of culture, the cells were starved by serum-free medium for 4 h before stimulated with 100 ng/mL EGF at 37 °C. Protein localization was detected by GFP and mCherry fluorescence using DeltaVision RT system (Applied Precision) with 60x1.4 NA oil immersion objective lens. After deconvolution using Softworx (Applied Precision), images were exported as 24 bit RGB images and analyzed using Image J software. The scale bars are 10 μ m. For corresponding live-cell imaging movie, merged CD2AP-GFP and mCherry-STS1 image were shown. The scale bars for the movie is 5 μ m.

Rebuttal Figure 2. Density-coloring volcano plots for the presented volcano plots in the manuscript. a. Related to **Fig. 2b-c**. **b.** Related to **Supplementary Fig. 5a-b**. **c.** Related to **Supplementary Fig. 5d-e**. **d.** Related to **Supplementary Fig. 5g-h**. **e.** Related to **Supplementary Fig. 5k-l**. **f.** Related to **Fig. 3b-c** and **Supplementary Fig. 6a**. **g.** Related to **Fig. 3d** and **Supplementary Fig. 6b-c**.

REVIEWERS' COMMENTS

Reviewer #1 (Remarks to the Author):

Since the revised manuscript has been considerably improved with additional results (e.g., DFT calculation with explanation, mass detection of the modified peptides), this reviewer agree to accept this manuscript in Nature Communications.